# A Usable Encryption Solution for File-Based Geospatial Data within a Database File System

Pankajeshwara Sharma [1,*,†], Michael Govorov [2,†] and Michael Martin [2,*]

1 School of Environment, University of Auckland, Auckland 1010, New Zealand
2 Department of Geography, Vancouver Island University, Nanaimo, BC V9R 5S5, Canada; michael.govorov@viu.ca
* Correspondence: pankaj.sharma@auckland.ac.nz (P.S.); michael.martin@auckland.ac.nz (M.M.)
† These authors contributed equally to this work.

**Abstract:** Developing a security solution for spatial files within today's enterprise Geographical Information System (GIS) that is also usable presents a multifaceted challenge. These files exist in "data silos" of different file server types, resulting in limited collaboration and increased vulnerability. While cloud-based data storage offers many benefits, the associated security concerns have limited its uptake in GIS, making it crucial to explore comparable alternative security solutions that can be deployed on-premise and are also usable. This paper introduces a reasonably usable security solution for spatial files within collaborative enterprise GIS. We explore a Database File System (DBFS) as a potential repository to consolidate and manage spatial files based on its enterprise document management capabilities and security features inherited from the underlying legacy DBMS. These files are protected using the Advanced Encryption Standard (AES) algorithm with practical encryption times of 8 MB per second. The final part focuses on an automated encryption solution with schemes for single- and multi-user files that is compatible with various GIS programs and protocol services. Usability testing is carried out to assess the solution's usability and focuses on effectiveness, efficiency, and user satisfaction, with the results demonstrating its usability based on the minimal changes it makes to how users work in a collaborative enterprise GIS environment. The solution furnishes a viable means for consolidating and protecting spatial files with various formats at the storage layer within enterprise GIS.

**Keywords:** spatial file security; usability; database file system; encryption; ECMSDK; multi-user files

## 1. Introduction

Unavailability or compromise of spatial data can have serious consequences. GIS data are a key component of decision-making in high-risk scenarios such as humanitarian and natural disasters. Timely access to relevant spatial data is of the utmost importance under these conditions; however, it is also important to understand that although there is a great need for data to be highly available, keeping this data *secure* is critical, as recently highlighted by researchers [1–6]. Almost 60–80% of all generated data have some geographic component to them [7], and, adding to this, the complexities associated with spatial data have increased dramatically [8]. Due to the multiplicity of available spatial formats, files are often scattered on different servers, namely, file, web, email, and document management servers [9]. Spatial "*data silos*" housing file data across multiple formats and the associated physical and virtual mediums reduce their utility and availability for critical decision-making scenarios [10–12]. These data silos also increase access points to sensitive or proprietary information.

The cloud helps eliminate technological barriers to collaboration and offers a equipped solution for connecting siloed data. It also addresses several limitations of traditional storage systems—supporting unstructured data models, high concurrency, low latency,

high flexibility, high scalability, and availability. Geospatial data are also increasingly stored in third-party services (such as those provided by *Google, Amazon,* and *Microsoft*) due to the cost and reliability of cloud solutions. Indeed, cloud-based geospatial data protection approaches now include encryption in their solutions. Esri's ArcGIS Online [13], Microsoft Azure Orbital's offering for geospatial data [14], and Google Bigquery [15] all offer solutions for geospatial data encryption at present. However, moving data storage infrastructures from on-premise to distributed cloud-based architectures increases security and privacy risks [16–18], and sensitive or valuable geospatial data must be safeguarded from companies seeking profit.

A cloud service provider guarding valuable data or digital assets must be *trusted*. Yet, in the past four years, prominent companies like *Facebook*, *Alibaba*, and *LinkedIn*, as well as a GIS company, *PeopleGIS*, experienced cloud security breaches, exposing millions of records, the majority of which contained location information [19,20]. While these breaches were from unknown adversaries, individuals within corporations can also gain unauthorized data access. Thus, spatial data security and the privacy of users in the cloud are still challenges [21], and the majority of organisations *do not* store mission-critical data in the cloud, as they argue there is a higher degree of confidence in security when the data are stored *on-site* [22]. As such, protecting file-based spatial data requires *revisiting* and *extending* existing storage security technologies that can be leveraged on-site.

Leading DBMS vendors (Oracle, IBM, Lotus, and Microsoft) came up with an innovative idea some years back—*creating a file system around a database*. This scheme of storing entire OS files in a database, which we call a *Database File System* (DBFS), bundled with commonly used protocol services (for example, web, shared folders, or file transfer), opens up several possibilities for spatial files. First, files previously existing in various data silos around the GIS enterprise can be consolidated. Second, this scheme promises database features that until now have only benefited structured data, such as content management, centralised authentication and access control mechanisms, which are beyond the capabilities of standard file systems. Finally, it offers robust DBMS management features to spatial files, namely, *reliability, user management, performance, scalability,* and *security*.

Cryptography is vital to prevent access to sensitive geospatial data within an enterprise in case of security breaches, ransomware attacks, and unauthorized access [23]. This protection is essential for complying with the recent legal and ethical regulations of different jurisdictions [24]. Many encryption solutions have been proposed for spatial data stored in traditional file systems and database management systems [25]. However, building a cryptographic solution is not about implementing encryption alone, as the associated challenges need to be addressed, namely, concurrency, key management, and ensuring solution usability through various GIS applications and protocol services. According to the literature, efforts to integrate security with usability often focus on enhancing the openness of security processes rather than making the system itself more useful [26].

*Usability* and *security* go together [27,28]. Most users tend to skip security applications or fail to incorporate security into their work due to encryption performance or complicated solutions that prevent them from working effectively in a collaborative environment [29]. Essentially, existing solutions for spatial file security cannot be integrated without affecting *collaboration* and cannot be *integrated* with the various GIS applications and protocol services. The root of these problems lies in security software internals: it is almost impossible to find a solution that deals with security issues in a transparent way, such that the GIS user needs only to know that the system has been secured; this is clearly a bad solution. A good GIS cryptographic solution's design must go further than the implementation of new ciphers and protocols into the realm of practical, *usable* security.

Considering these concerns, a localised spatial file storage infrastructure is needed to address traditional file systems' limitations by furnishing consolidated storage and management, including content management features, for files in various formats while protecting them at the storage layer from malicious insiders and external threats. Nevertheless, the key challenge is to ensure that any such solution is *usable*. Such a solution will enable access

from a central repository, resulting in improved GIS collaboration and decision-making capacity and assisting with fulfilling data privacy laws and regulations.

This paper proposes a DBFS as an alternative consolidated storage platform for spatial files. *Cryptography* is applied for added protection, addressing several key challenges during the cryptographic design process, namely, the choice of encryption algorithm, providing for multiple users, concurrency, automation, and key management methods. Thereafter, a GIS security model is presented, and the developed solution is evaluated against various adversarial attacks. We include usability testing to demonstrate that the encryption–decryption solution accommodates the aforementioned intricacies. The test focuses on the implementation of the solution within contemporary enterprise GIS services using file shares and the QGIS desktop application and highlights the *security, concurrency,* and *seamless integration* aspects. Another implementation demonstrates how the solution can be used to provide additional security for spatial files stored in a database-centric GIS Portal and geodatabases within a Web Atlas.

The rest of this paper is organised as follows. Section 2 presents related works in the area and the research objectives. Section 3 introduces the architecture of the encryption and decryption solution for spatial files, and Section 4 describes the automation of the solution. Section 5 presents the usability analysis and implementation of the encryption solution. This is followed by a discussion of the security and usability of the solution in Section 6. A discussion of the findings is included in Section 6, and the conclusion is in Section 7.

## 2. Security and Usability Concerns for Spatial Files within Enterprise GIS

While enterprise GIS stores much of its geospatial data in a database, there is still much confidential data in file formats, for instance, shapefiles, geodatabases, and geoJSON files. For the storage and management of these files, traditional file servers have been extensively used in GIS. They provide enterprise users with an easy-to-use interface and adequate security features; however, they can support a limited number of users and lack the *content management, reliability, scalability, availability,* and *security* features required to manage spatial files in collaborative enterprise GIS environments. As a result, spatial organisations often store files on several file servers, document management systems, web servers, and so on, which affects information access since there is no central repository. In such a data-siloed scenario, applying encryption only adds more complications for users.

Traditionally, management and sharing of the most sensitive geospatial data are tasked with on-premise databases and data stores. For further data protection, cryptography is applied during storage using a solution such as Bitlocker on a hard drive, an encrypted geometry column in a database, or over the network using encryption such as SSL [30,31]. The encryption approach proposed by Li [32] secures conventional database-based spatial data within an Oracle spatial database. Approaches for protecting spatial files during storage and transfer include those focusing on vector-based files [25,33–37] and those focusing on general spatial files [31]. For instance, Dakroury et al. [31] propose a combination of encryption and digital watermarking techniques, while Ghaleb et al. [38] propose an encryption module for the QGIS software to prevent users from making unauthorised copies of the original data. While these approaches have focused on specific challenges, they are limited in providing an overall encryption solution that *consolidates* the different spatial file types, further *safeguards* them using a robust access control mechanism, and provides a *usable* encryption solution for enterprise GIS.

In recent years, technologies and solutions in virtual storage have emerged, and many of these have been widely used by Internet platforms. For geospatial data, there is also a need to *evolve* traditional file systems and relational database solutions to a distributed, virtual, and software-defined storage system so that storage scalability and processing capability can meet future challenges. Current Big Data integrations on spatial data have mainly focused on structured data and have ignored unstructured (for example, file-based) data [39,40]. Additionally, as Big Data management is just in its infant stage [41], many

of the NoSQL database systems lack encryption mechanisms that support security for data-at-rest and data-in-transit.

Cloud-based GI infrastructure and services may be the future of GIS. However, their service providers can track and exert power over data, resulting in rightful custodians relinquishing control [42]. The prominent companies are growing based on the support of the US government [43] and can access data irrespective of its origin using their globally placed servers. This has not only raised concerns around offshore data security and privacy but also the legal and privacy frameworks that the data are subject to. The *MapSafe* browser-based application [44] and the recent integration of its features in GeoNode [45], permit sovereign data owners to carry out geomasking, encryption, and blockchain notarisation security functions from within the browser by themselves. This furnishes a complete data sovereignty application for safeguarding and verifying geospatial datasets when shared via the cloud without reliance on a third party. While this approach protects geospatial files when shared outside the GIS organisation, more needs to be done to consolidate and protect geospatial file-based data within the organisation.

Another approach is by Zhang et al. [46], for which spatial files are encrypted before storage on a decentralised file system, namely the *Inter Planetary File System* (IPFS). However, common GIS applications *cannot* be integrated to use IPFS, since it is not a shared server platform where multiple users can have both read and write access to shared folders. Therefore, a security solution is needed that can be deployed *on-premise* within the enterprise and is *usable* with various commonly used GIS applications. Once its security is guaranteed, this model could be applied to distributed file systems such as IPFS.

Not long ago, commercial DBMS vendors began offering file system products that enable users to easily store and access files within the database. *Database File System* (DBFS) products such as the IBM DB2 Content Manager [47] and the Oracle Content Management Standard Development Kit (CMSDK) [48] use the database to store OS files, replacing many complex tasks usually performed by an OS. This innovative architecture for the first time opens up several opportunities for spatial files that are not currently available with traditional file systems. It raises the possibility for any spatial file to be *created*, *reviewed*, *corrected*, *approved*, and finally *published*, with appropriate access restrictions for user groups or simple users of the DBMS [48]. At the same time, the bundled protocol servers (for example, those with the CMSDK) allow the DBFS users to access and work with spatial files using the most commonly used GIS protocol services [48]. By using the DBFS, the spatial organisation can be confident that content is secure and accessible from a central location. Geospatial researchers have previously highlighted these advantages while proposing CMSDK for spatial file storage within Web GIS [49]. We are continuing in that line of investigation by investigating a usable encryption solution built around the DBFS for spatial end users. Note that Oracle released the product as open source in 2014 [50], and it is now known as the *Enterprise Content Management Standard Development Kit* (ECMSDK). The product is available from www.ecmsdk.com (accessed on 18 April 2024) and has been used for all investigations in this study. The product mounts an additional logical drive on the computer, and files placed on this drive are stored in the Oracle database.

Adding a database layer on top of the OS file system may affect the performance of file retrieval, storage, and editing. However, such a facility built on a relational DBMS model can facilitate the critical storage requirements for spatial files, whereas traditional file systems cannot. Moreover, DBMSs provide a robust enterprise solution as they are actually more scalable than file systems [48], support an enormous amount of data, perform reasonably well under severe loads in a multi-user environment, and provide database clustering and standby database features. This can be a storage infrastructure solution that spatial file *management* and *security* currently require today when spatial file content is beginning to be managed in so many diverse ways.

Storing OS files within a database can be overkill—something that may have impacted the uptake of DBFSs since their inception. In this regard, geospatial researchers have previously investigated the feasibility of using a DBFS for spatial file storage, management,

and security. Govorov et al. [51,52] investigated the performance of *input, retrieval,* and *updating* spatial files using MapInfo software and found a difference of about 1–2 s for processing spatial files (with sizes up to 100 MB) by using CMSDK storage compared to the native Windows filesystem. Further *encryption* for spatial files within the DBFS presented reasonable times for small-sized spatial files using traditional algorithms. Security solutions, however, are not about encryption *alone*, and a limitation of that work, and also of other spatial file encryption solutions in the literature, is the lack of an overall encryption–decryption solution that caters to file encryption in a *multi-user* environment.

A security system needs to be *usable*. Mechanisms implemented for security and privacy *should not* complicate workflows and must be *transparent* for the user [53]. However, increased usability of software applications generally reduces security and privacy up to a significant level [54]. Conversely, solutions focusing on security and privacy usually lead to decreased usability and use, as has been well documented in the Information Systems domain [53,54]. While establishing a trade-off between usability and security, complexity should be avoided during user interface design [55]. Similarly, a GIS is not an exception, and only a few users will ever use real security applications that take too long or are too complicated and present too many details of the cryptographic process. While GIS application security against usability is a relatively unexplored topic [56], researchers focusing on the usability aspects of GIS applications have highlighted that user-friendly and easy-to-use interfaces were most important. Komarkova et.al. emphasised that users need to *"access spatial information quickly, without any special software tool and without any special training"* ([57], p. 1). Our study is focused on addressing this challenge.

The notion of usability often revolves around metrics of *effectiveness, efficiency*, and *user satisfaction* [58,59]. The International Standards Organisation (ISO) formally defines usability concerning information technology as *"the extent to which a product, system, or service can be used by specified users to achieve specified goals with effectiveness, efficiency, and satisfaction in a specified context of use"* [60]. Relating to the definitions in ISO 9241-11:2018 [60]:

- *Effectiveness* concerns the completeness and accuracy when users carry out the tasks when using a product;
- *Efficiency* relates to the resource expenditure; and
- *Satisfaction* is about the user's attitude regarding the use of a product.

Considering these metrics, we explore an alternative security solution that offers a centralised storage repository for managing and securing spatial files of various types, and also protects these files at the storage level without changing the way GIS end-users work with data server repositories. Achieving this functionality would require multiple technological components to be integrated, and investigating the security and usability of the overall approach is part of our objectives, which are as follows:

- To be able to store heterogeneous spatial data files (json, shapefile, gdb, and so on) in a container that indexes everything spatially;
- To implement file encryption within the storage repository, resulting in a usable encryption solution for use with various GIS applications and user types; and
- To investigate the usability of such a solution in terms of its *effectiveness, efficiency*, and *satisfaction*.

In summary, current GIS storage mechanisms are limited in consolidating different spatial file formats and in providing a usable encryption–decryption solution. Securing these files can play an important role in enterprise GIS environments, as current solutions and systems do not address this concern. The latest desktop GIS applications (such as ESRI ArcGIS Pro and so on) do not address the security solution at all for working with heterogeneous files and personal geodatabase workspaces. Available technologies today can be combined to facilitate these security limitations. This research examines the feasibility of storing spatial files in DBFS and applying cryptography for their further protection. Such a solution has the potential to be applied in other domains; however, the main relevance to

enterprise GIS is addressing data silos and the resulting security vulnerabilities through consolidating spatial files into a single repository.

## 3. Proposed Security Solution for Spatial Files

Our approach to protecting valuable spatial files in enterprise GIS environments comprises multiple components. Figure 1 shows the proposed storage security solution in the form of a spatial end-user interaction model in an enterprise GIS environment. Spatial users, using any protocol server, are first authenticated against their ECMSDK accounts. After a successful login, each file is decrypted. Access to spatial folders and files is controlled with the DBFS *access control mechanism*: that is, the ECMSDK *Access Control Model* (ACM). Upon user log out, files are encrypted for protection. User credentials can be protected in transit using SSL encryption, which can be integrated with ECMSDK [50].

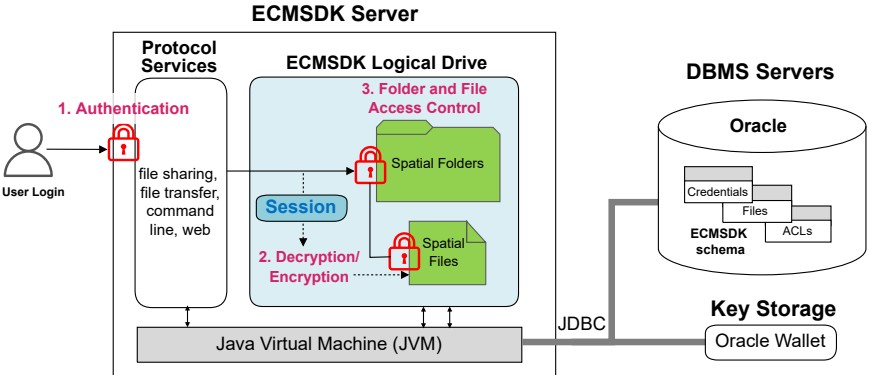

**Figure 1.** The user interaction model of the storage security solution. GIS users can store sensitive spatial files with various formats centrally within the database and can protect them using authentication, authorisation, and encryption without changing how they work with existing GIS applications.

### 3.1. A DBFS for Consolidation of File-Based Spatial Data

A centralised repository provided by a DBFS model delivers three key benefits to spatial organisations: *First,* since a DBFS provides storage for various file formats, namely *documents, web pages, e-mails, and so on,* and ships with various integrated, commonly used protocol servers for accessing these data, it *consolidates* spatial files previously stored on separate machines managed by separate server applications into a single DBFS repository, which spatial end-users can access through familiar interfaces. This way, spatial organisations can preserve their investments in existing spatial datasets in numerous file formats and significantly reduce the number of integration interfaces GIS users and applications have to make.

*Second*, since the DBFS is built on top of a general-purpose database to store and manage files, several features of the database and application server are integrated into the DBFS, providing an alternative to the limitations of traditional standard file systems. ECMSDK, for instance, integrates with the *reliability, disaster recovery, read consistency, replication, scalability,* and *availability* features provided by the Oracle database [61]. Additionally, for spatial administrators, a single system for file storage and access means single-system management. They only have to maintain, administer, backup, and upgrade a single system, which also simplifies disaster recovery planning [62].

*Third*, a DBFS such as ECMSDK makes use of the relational nature of the database to build upon and provides several content management features missing from traditional file systems that can be used to help business workflow processes in spatial organisations. These important content management features for files include *indexing and searching based on custom attributes, versioning, check-out and check-in operations, change notification, automatic expiration,* and *locking* [48].

To complement these benefits, a DBFS such as ECMSDK offers a powerful and multi-layered security model. Since spatial files are stored inside the database; database security

ensures non-database users cannot access spatial files. For legitimate users, ECMSDK ensures proper user authentication using an internal credential manager or, more recently, with an LDAP using the *Oracle Internet Directory* (OID) [62]. Thereafter, ECMSDK allows user actions to be controlled consistently and logically with its access control model [48]. Since access control is a function enforced from within the database repository rather than protocol servers, it is enforced consistently, regardless of which protocols are used to access files. In addition, a DBFS such as ECMSDK provides full support for Internet security standards, e.g., *SSL* and *firewalls*, as well as protection against harmful programs, such as viruses [48]. When these features are combined into a single repository, organisations can be confident that valuable spatial files are managed, searchable, and secure at a central location—something not offered by traditional standard file systems.

**Access Control Model (ACM):** ECMSDK enforces proper authentication and authorisation mechanism. Users first require a login, and afterwards, their privileges are tightly controlled on files and folders through an associated *Access Control List (ACL)* that contain users/groups and their privileges. From our investigation into the product's security model, the authorisation controls of the particular OS are incorporated with ECMSDK. Therefore, secure access is double-checked for additional security. After successful authentication, OS users with a corresponding account with ECMSDK are authorised under these accounts and have related privileges. *Non-ECMSDK users* (including the OS administrator) are represented as guest users and have guest privileges only. Moreover, the combination of ECMSDK and OS access control settings for shared folders provides an additional security layer to folders, as when these two access controls are combined, the more restrictive permissions of the two are the user's effective permissions.

### 3.2. Encryption Challenges for Files within the DBMS

Here, we outline the several challenges that were faced while implementing the encryption of files within a DBMS. These relate to how encryption algorithms are implemented, how DBMS handle large data, and how to obtain optimum encryption performance.

**Reducing physical read–write operations:** Reduction of data transfer and lookup is a critical component of database optimization and has impacts on the realized speed and performance of a database. Ideally, database read/writes are kept to a minimum as, in a practical sense, this results in a more seamless experience for the user whereby only the data that are needed are loaded and written. To reduce expansive disk I/O read/writes, *more* data should be read in each read–write operation. However, this is limited by two factors. The first is the *maximum* allowable data storage in a scripting language: in our case, the `VARCHAR2` datatype in Oracle PL/SQL is limited to 32,767 characters (32 Kb) [63]. The second is the *amount* of input bytes that an encryption algorithm allows: that is, the AES encryption algorithms provided by Oracle within the `DBMS_CRYPTO_TOOLKIT` [64]. The amount of information that can be read by the AES encryption mechanism is a critical element of security. The algorithm parses data in 32 Kb chunks to limit the potential of data leakage in the case of errant data being exfiltrated in the process of transfer. *However*, with the small size of the transfer, this increases the number of calls that are required to ensure the user has the information he/she needs at the time of usage. Increasing this to a larger amount reduces physical I/O read/writes.

**Encrypting files larger than the database buffer cache:** The DMBS server returns an error when encrypting files larger than the Oracle DB cache size, indicating no more *free blocks* left in the DB cache. For GIS, the cache size matters. Point clouds, high-resolution satellite data, and other large data formats often require dynamic data streaming between the hard disk, database servers, and main memory (RAM) in the user's computer. Management of this is a complex dance, and cache size is a prime determinant of the size the stream can stage between access reads and writes. One could increase the *DB buffer cache* size to a very large value, and this should not be an issue, as in DBMS today, servers with 128 GB RAM (memory) are a common solution. However, GIS data are now often large enough to outpace the available cache size. Thus, in our approach, *dirty buffers* are written

to disk regularly by encrypting each file in large chunks and issuing the `COMMIT` statement multiple times during encryption or decryption.

**Excessive redo log generation:** Traditionally, for any changes made to the database, DBMSs create new entries in the *redo log files* for recovery purposes. For large spatial files, this raises excessive logging overheads in the form of additional CPU time and disk space usage. To improve encryption performance, the `NOLOGGING` setting for *Oracle Large Object* (LOB) can be used.

**Searching encrypted file data:** Core organisational functions require searching information that has been encrypted. Differential privacy focuses on allowing users to query and retrieve aggregate statistics from spatial data without viewing individual data points, and it is possible that we can integrate these algorithms (instead of AES) in the future if they become open-source and their security has been tested and proven by the research community. Nevertheless, ECMSDK allows custom attribute (*metadata*) to be defined and associated with files, which can be used for searching. Since encryption only encrypts the file content, these custom attributes can be used for searching, preventing the need to search the encrypted content itself.

*3.3. Strong Encryption for Spatial Files*

Ensuring *end-to-end* security for spatial files at the storage layer involves protection when the DBFS *access control model* has been circumvented and also when these data are transferred onto other removable media for backup, destruction, and so on, and lie around unprotected. The field of Information Systems proposes six primary mechanisms for stored data encryption, with their respective advantages and disadvantages [65–67]:

*(a) The Application-Integrated* approach encrypts data at the application layer: that is, from within the application. While this is secure, all GIS client applications must be modified to support the encryption–decryption model. *(b) For DBMS-integrated* encryption, entire database datafiles are encrypted, adding needless overhead. *(c) DBMS-based* encryption encrypts only selected pieces of information. In our case, however, this approach has to be workable with various GIS applications. *(d) Operating-System-based* encryption requires third-party encryption software for encryption, which could be difficult to integrate with GIS client applications. *(e) Operating-System-integrated* encryption allows end users to encrypt files, folders, or file systems on different operating systems: however, the solution requires the encryption of entire database datafiles or the file system. *(f) Encryption-server-based* approaches handle all encryption processing: however, they also requires modification of the GIS client applications. *(g) Storage-level* encryption encrypts data at the storage subsystem, again adding unnecessary overhead.

*DBMS-based* encryption was chosen, as this approach only encrypts the spatial files stored within the DBFS database, preventing unnecessary overhead with regard to encryption of the entirety of the database datafiles. While this is similar to database record-level encryption, it differs through the need for pragmatic implementation, especially with the various GIS applications and protocol services. Since both encryption and decryption processes will be carried out within the DBMS server used by the DBFS, we used *symmetric key* encryption where both processes can access the key, and in doing so, a public–private key approach was not needed. A public–private key approach would require GIS users to provide the private key for data decryption via a client application. In our automated approach, this would bring an extra step for users after login and require changes to client GIS applications to enable users to insert their private key. More details of the encryption scheme are described in the next section.

Current encryption algorithms, such as AES, provide an efficient way to safeguard spatial files. To recover plaintext using an AES algorithm with an effective key length of 256 bits would take the strongest supercomputers today longer than the lifetime of the universe to break it [68]. As such, using the AES algorithm will make it impossible to make use of any secret information obtained from compromised database datafiles (that contain the spatial files) from the storage layer. The second component of our security solution com-

prises using encryption for spatial files within the DBFS. Subsequently, we investigated the performance of the AES algorithm when encrypting spatial files within ECMSDK against encrypting spatial files stored directly on the OS (e.g., *Windows NTFS*). We investigated the spatial file encryption speed within ECMSDK using the `DBMS_CRYPTO_TOOLKIT` provided with Oracle 11g XE Database [69] and the *OpenSSL* encryption tool [70].

Table 1 compares encryption times in seconds for spatial files using the AES (256-bit) algorithm within ECMSDK and on Windows NTFS. In doing so, we carried out spatial file encryption and decryption in three scenarios: (a) within ECMSDK using Oracle `DBMS_CRYPTO_TOOLKIT`, (b) within ECMSDK using *OpenSSL*, and (c) within Windows NTFS using *OpenSSL*. The first two approaches compare file encryption and decryption performance within ECMSDK using the Oracle-provided `DBMS_CRYPTO_TOOLKIT` against the performance provided by the *OpenSSL* tool (a Windows executable that is called externally from our encryption scripts). The third approach compares the encryption of spatial files within ECMSDK versus those stored directly on the OS (*Windows NTFS*).

For the first approach, the time taken to encrypt an ECMSDK file was retrieved using encrypt and decrypt PL/SQL procedures using the SQLPlus tool. For the second and third approaches, times were retrieved using Windows batch scripts with OpenSSL encryption and decryption commands. Both these scripts and their usages are attached in Supplementary Materials online (https://github.com/sharmapn/DBFSFileCrypto, accessed on 18 April 2024). Note this GitHub repository also contains most other Supplementary Materials referred to in this article. All experiments were performed on a Dell Inspiron machine with an Intel 1.6 GHz i5-8265u processor and 8 GB RAM manufactured by Dell Technologies, sourced from Xiamen, China. The software utilised consisted of Oracle 11g Database Release 2 running on the Windows 10 OS. The ESRI shapefiles used in the investigation are sourced from publicly available datasets (https://mapcruzin.com/download-free-arcgis-shapefiles.htm (accessed on 25 April 2024)).

**Table 1.** AES encryption and decryption times (in seconds) for shapefiles within ECMSDK and on Windows filesystem.

| Spatial File | Filesize (Mb) | DBMS_CRYPTO on ECMSDK | | OpenSSL on ECMSDK | | OpenSSL on NTFS | |
|---|---|---|---|---|---|---|---|
| | | *Enc.* | *Dec.* | *Enc.* | *Dec.* | *Enc.* | *Dec.* |
| france-points.shp | 1.50 | 1.13 | 1.07 | 0.48 | 0.17 | 0.04 | 0.04 |
| france-waterways.shp | 6.32 | 4.73 | 4.49 | 1.02 | 0.29 | 0.06 | 0.05 |
| france-natural.shp | 11.67 | 8.74 | 8.30 | 1.62 | 1.75 | 0.11 | 0.07 |
| indonesia-natural.shp | 11.95 | 8.95 | 8.50 | 1.65 | 0.55 | 0.07 | 0.07 |
| germany-points.shp | 13.89 | 10.41 | 9.88 | 1.87 | 0.92 | 0.07 | 0.14 |
| britain-waterways.shp | 16.12 | 12.08 | 11.46 | 2.12 | 1.31 | 0.08 | 0.09 |
| china-buildings.shp | 26.53 | 19.87 | 18.87 | 3.29 | 1.90 | 0.12 | 0.34 |
| india-natural.shp | 32.13 | 24.07 | 22.85 | 3.92 | 2.38 | 0.13 | 0.37 |
| china-natural.shp | 33.75 | 25.28 | 24.00 | 4.11 | 2.68 | 0.14 | 0.48 |
| india-waterways.shp | 36.21 | 27.13 | 25.75 | 4.38 | 2.77 | 0.15 | 0.56 |

The encryption times for spatial files using the AES algorithm using all three approaches have a linear correlation to file size. The version of the algorithm implemented in `DBMS_CRYPTO_TOOLKIT` is approximately 6.6 times slower than when using the OpenSSL tool for encrypting spatial files in ECMSDK, which encrypts and decrypts 1 Mb of file size in approximately 0.15 s (or an average of 8.74 Mb per second) using a 32-character key. Using the OpenSSL tool (version 3.0.1) on ECMSDK with different key lengths results in a small time difference: for example, the average difference in time between 16- and 24-byte keys is about 8%, but the time difference between 24- and 32-byte keys is about 1%.

However, using OpenSSL on ECMSDK is still much slower than using OpenSSL to encrypt files stored directly on Windows NTFS—on average, it takes 15 times longer. This is due to the intermediary database layer between the encryption tool and the files.

Nevertheless, the AES algorithm implemented in the OpenSSL tool provides satisfactory encryption–decryption performance. The results are quite good, as an average time of 8 Mb per second is quite reasonable and can be used practically for encrypting spatial files. For encryption and decryption of files in ECMSDK, the OpenSSL tool can be easily called externally using Oracle PL/SQL from within our scripts. The average OpenSSL AES encryption speed is about 0.15 s per megabyte, or about 2.5 min to encrypt or decrypt a 1 GB spatial file on a desktop machine (*1.6 GHz Intel i5-8265U* CPU with *8 GB* RAM). In our solution, the total computational cost is only the time required to complete the encryption–decryption operations shown in Table 1. Other than this, no additional overhead is associated with this scheme since the triggers that call these procedures execute instantaneously (see Section 4.4).

These results can be improved. As encryption and decryption were carried out on a standard desktop, an enterprise server with a high-end multiple-CPU architecture and more RAM (for larger DB_CACHE_SIZE) could reduce the encryption speed drastically, at least by 20 times, which would be more suitable for enterprise GIS. Additionally, multi-processor UNIX servers can be used so that large files are split and then encrypted/decrypted simultaneously by multiple processors. This way, the resulting encryption time can be decreased, at least by multiples of the number of processors. Since AES is a very well-studied encryption algorithm and it presents us with reasonable encryption times for spatial files within a DBFS, we recommended its use. Selecting a spatial file security solution based on AES presents a safe and wise decision as it provides the best encryption security, regulatory coverage, and position for future development.

## 4. Automated and Transparent Encryption and Decryption

This section presents schemes for *single-user* and *multi-user* spatial files, an *automated* encryption and decryption model, and a *key storage* scheme, which are part of our proposed solution. Figure 2 outlines a schematic model of the proposed storage security solution for spatial files built around a DBFS. In a typical enterprise GIS environment, two types of files need protection during storage: single-user and multi-user files. The first set belongs to individual end-users and requires access by an end-user only. The second set requires sharing between more than one end-user at a project level and does not necessarily belong to any particular end-user. Our solution includes encryption schemes for both sets.

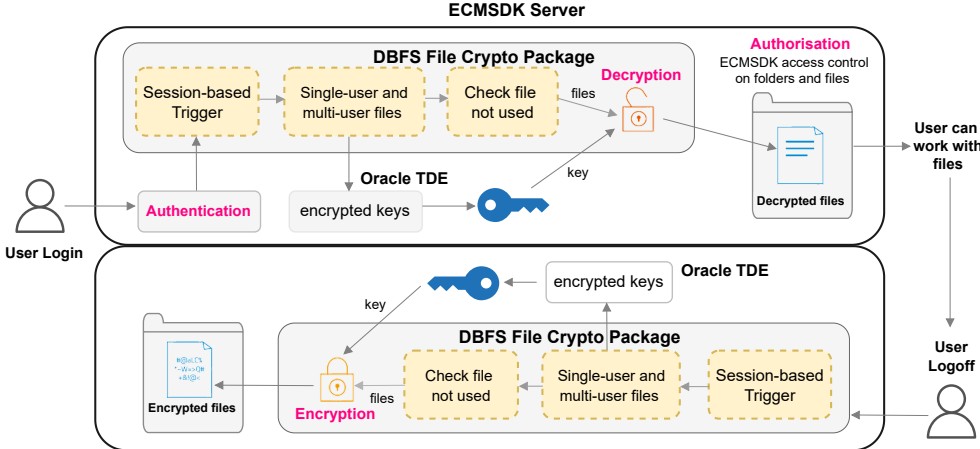

**Figure 2.** Schematic model of the security solution for spatial files. As illustrated in the top region, GIS users are first authenticated against their DBFS account, and thereafter, upon session-creation, their single- and multi-user files are decrypted. Thereafter, access to files is controlled via the DBFS access control model. The bottom region illustrates the encryption process upon user logoff.

### 4.1. Single-User Spatial Files

The first set of sensitive spatial files that need protection are *single-user* files. These files are created and owned by spatial end users and should not be shared with others. They are normally stored on users' computers or in their home folder within the file server. Using the proposed security solution, these single-user spatial files are stored in their home directories on the DBFS server. Since this set of files is controlled by end users, the encryption scheme must allow users to define which files are to be encrypted for protection, without the involvement of administrators. Ideally, on user logon, these files must be automatically decrypted for use and automatically encrypted at logoff. This automation is described in Section 4.4 shortly.

The specification of a single-user sensitive spatial file to be encrypted has been designed to be simple and can be performed using any GIS application or protocol service used to connect to ECMSDK shared folders, that is, from a desktop, web, or mobile device without the involvement of administrators. The proposed implementation system used here is to adopt the *.enc* file type. Use of this file type is compatible with existing file formats such as *\*.shp* and *\*.json* and does not add additional file overhead. The *.enc* file extension is added to the end of the filename, for example, *"historical_places.shp.enc"*, and the extension indicates the files are currently locked for protection and require decryption. For encryption of file-based geospatial layers, only files with valuable data would require this extension. For example, in the shapefile format, only *.shp, .dbf,* and *.shx* could have the *.enc* format appended. Figure 3 demonstrates a GIS user's single-user files decrypted for use, with the full user interaction workflow are included in Supplementary Materials online, and this YouTube video https://www.youtube.com/watch?v=zMJV5zV0-zs (accessed on 18 April 2024).

After decrypting, and using the file, if the file does not need to be encrypted, then users can simply retain the original file extension and it will not be encrypted. Since this set of files has an ECMSDK *Access Control List* (ACL) that controls access, no other DBFS users can change the file extensions, delete the file, or view its contents while they are in decrypted form on the server. To prevent the entire encryption scheme from being dependent on one set of keys, each user has a unique set of encryption keys that are used for both encrypting and decrypting the user's files. A database table created within the database administrator schema is used to store encryption and decryption keys for each user's files in order to prevent access by other database users (see Supplementary Materials).

### 4.2. Multi-User Spatial Files

The second set of sensitive spatial files is *multi-user* files. These files are shared between multiple users, for instance, during projects, via shared folders from within an organisation's file servers. In our scheme, these files are kept in separate "project" folders within the DBFS. Designing an encryption scheme for multi-user files is a relatively challenging task, as it requires maintaining a list of files to which not only one particular spatial DBFS user should have access but also other privileged users. It is also necessary to make sure that files are not encrypted while users who should have access to them have active sessions in the DBFS. Figure 4 demonstrates this aspect, where, upon logoff, not all of a GIS user's (Scott's) multi-user files are encrypted since another privileged user (Alan) has an active session. The full user interaction workflow is included as part of Supplementary Materials. Not all files that users are granted access to using ECMSDK ACM need encryption. Therefore, the multi-user encryption scheme is implemented independently of the user's access privileges on ECMSDK files.

The scheme is implemented using a catalogue that is used to determine the spatial users and the corresponding multi-user files that should be decrypted for their use and encrypted after their use. The catalogue is implemented using a database table, and the specification of multi-user GIS files and granting privileges to users in this catalogue is to be handled by the administrator. All users and the directories containing the files (for example, *"O:\home\scott\historical_places.SHP"*) that need to be decrypted for use

and encrypted back afterwards are stored in tables (see Supplementary Materials). Since single-user files might be made multi-user as soon as someone else needs to collaborate on a project, to allow multi-user access, these files can be moved into the project folders, and their access settings can be changed (using ACLs) correspondingly.

### 4.3. Access to Decrypted Data

In our proposed solution, the original encrypted Large Objects (LOBs) containing the spatial files **are never replaced** with a decrypted version. During decryption, instead of replacing the original encrypted LOBs in the database's permanent tablespace, the LOBs are decrypted and stored in the temporary tablespace. Oracle DBMS supports "*temporary tablespaces that contain data that persists only for the duration of a user's session*" [71,72]. This permits temporary LOB datatype storage in such a way that data are not permanently stored in the database. Thus, we replace the LOB locator pointing to the original spatial file with the temporary locator that is stored in a separate ECMSDK table within the temporary tablespace. After decryption, users access, use, and update this copy of the LOB that contains the file content.

After users log off, these spatial files stored as temporary LOB content are encrypted back, replacing the original LOB in the permanent tablespace. For authorised users, read and write access to encrypted and decrypted files within ECMSDK is controlled through setting its *access control list* (see Section 3.1). The decrypted files within the temporary LOBs will be freed from the temporary tablespace as soon as the database crashes. **This way, an adversary with access to the disk will always have access to only the encrypted spatial files.** Using temporary LOBs in the decryption process does not require any additional overhead, as instead of data being written into the same file's LOB object, it is written into the temporary LOB object.

### 4.4. Automated Encryption and Decryption Model

The security solution allows spatial end-users to use the encryption facility with minimal changes to how they used to work previously with their GIS and applications. This consists of an automated model that allows encryption–decryption processing to be performed in the background and hidden from the spatial user.

*Session-Based Triggers:* To perform decryption and encryption procedures automatically when a spatial user accesses or exits the DBFS, a *session-based* trigger based on the ECMSDK user sessions' `ODMZ_SESSION` table is used, that calls the single-user and multi-user file decryption and encryption procedures upon user login and logoff, respectively. This way, files are decrypted and encrypted automatically, without any change to how users interact with file shares using GIS applications.

*Concurrent User Sessions:* An encryption solution must also keep track of the active user sessions to prevent *double-encryption* and *double-decryption*, which would make the files unusable for other privileged users still using them (have active sessions). If certain multi-user files are already decrypted for a privileged user (*user A*), another privileged user's session creation would call the decryption again, overwriting the temporary LOB with the original encrypted file content *again*, and by doing so, would overwrite the modifications made by *user A*. Similarly, upon another privileged user's session termination, encryption should be avoided, as this would make the LOB point to the one in the permanent tablespace containing the encrypted file content, making the file unusable to other privileged users who still have active sessions, while their file changes would be lost. To allow other privileged users to keep using the *multi-user* spatial files, when one privileged user creates or terminates a DBFS user session, the session-based trigger calls the `check_in_use()` procedure that ensures there are no active sessions from all intended users who should have access to these files only after which decryption or encryption procedures are executed.

The entire proposed encryption–decryption solution for single-user and multi-user spatial files is implemented in PL/SQL and contained within the *DBFS File Crypto package*. The package, along with the session-based trigger, is available as part of Supplementary

Materials. The automated model and the encryption–decryption processes are directly connected, without any intermediary processes involved. As a result, the only performance concern when implementing the entire encryption system is the processing time of encryption and decryption, as shown in Table 1.

The entire decryption process executed upon login onto the DBFS for user's single- and multi-user files is encapsulated in Algorithm 1 next. The encryption process, included as part of Supplementary Materials, has similar steps.

---

**Algorithm 1:** Decryption process for single- and multi-user files within ECMSDK executed upon user login.

---

1   **Input:** User ID of DBFS user who has created a session
2   **Output:** Decrypted single- and multi-user files of user
3
4   **if** *DBFS user session created* **then**
5     Check for any existing sessions of the *user*
6     Exit if any sessions exist          `// do not proceed`
7   **end**
   `/* Check for files with .enc extension for single-user files      */`
8   **for** *singleUserFiles of user* **do**
9     **if** *file has .enc file extension* **then**
10       Retrieve file *decryptionkey* from a DB table   `// protected using Oracle Wallet`
11       Add *file* to *decryptionList*
12     **end**
13   **end**
   `/* Ensure no user has active sessions for multi-user files         */`
14   **for** *multiUserFiles of user* **do**
15     **if** *no other user with privilege on file has active sessions* **then**
16       Retrieve *decryptionkey* from a DB table     `// protected using Oracle Wallet`
17       Add *file* to *decryptionList*
18     **end**
19   **end**
   `/* Decrypt all files in the decryption list                       */`
20   **for** *files in decryptionList* **do**
21     Create a temporary *file* (LOB)
22     Decrypt the original *file* into the temporary *file*
23     Point original *file* reference to temporary *file* `// original file never decrypted`
24   **end**

---

### 4.5. Encryption Key Storage

The issue of passwords is one of the most researched topics in usable security [26]. In our approach, the keys stored in the database are further protected using the *Transparent Data Encryption* (TDE) facility provided by Oracle Database 11g Release 2 as part of the Oracle Advanced Security option [73]. The TDE option enables automatic encryption and decryption of database table columns (with small-scaled data), while the associated key is securely stored in an encrypted container called a wallet (*Oracle Wallet*). Using an external security module such as this wallet separates ordinary application functions from encryption operations, making it possible to separate duties between DBAs and security administrators [74]. Security is boosted since the DBA would not know the wallet password, as it would be necessary for the security administrator to input it.

Using this approach, the cryptographic keys for single-user and multi-user spatial files are stored in a table in encrypted form and are decrypted transparently using TDE when the decryption procedure queries the table. A description of this table as well as the SQL commands that the DBA can use to create or delete single-user keys for the DBFS users are part of Supplementary Materials online. The Oracle Wallet can be initialised by issuing the following command, which creates a wallet, creates a master key for the entire database, and opens the wallet: "`ALTER SYSTEM SET encryption key identified by 'password'`". To prevent the entire encryption scheme from being dependent on one set of keys, a unique set of encryption and decryption keys is used to encrypt and decrypt each user's files (stored in a database table).

## 5. Usability Analysis and Implementation of the Security Solution

This section focuses on the usability evaluation of the security solution. In doing so, we also demonstrate its use in a collaborative enterprise GIS environment. How the encryption solution can be implemented for protecting spatial files within a database-centric Web Portal [75] and geodatabases within a web atlas [76] are part of Supplementary Materials.

The design objective of the encryption–decryption solution was to ensure its usability. In this regard, *usable security* has received little attention from the developer community, for whom *usability* is considered entirely independent and less important than security [54]. Usability evaluations of GI web applications and portals have focused on evaluating efficiency, effectiveness, and user satisfaction. Kalantari et al. [77] carried out a spatial metadata usability evaluation of data directories and portals and reported several issues related to the aforementioned three aspects of metadata. He et al. [78] investigated the same three metrics when evaluating the usability of a geoportal for complying with industry specifications, reporting several weaknesses in its design. Therefore, we also focus on these three aspects for investigating our security solution's usability.

### 5.1. Usability Evaluation

Our approach is based on the ISO 9241-11:2018 framework [60], where the usability evaluation is split into three sub-scales, including *effectiveness*, *efficiency*, and *satisfaction*. The two co-authors, who have several years of experience in GIS, carried out the evaluation. To this end, the tests correspond to *Non-Technical Single User Testing* [79].

**Usability testing** was used to measure the security solution's usability. The two respondents were given ECMSDK user accounts named Scott and Alan and were asked to operate ten tasks twice (see Table 2). The time spent completing the tasks was documented. The respondents' failure to carry out the tasks was also recorded, whether it was due to them or the solution. Since using the solution is the same as using traditional file systems (except the part where users specify the files to be encrypted or decrypted), it incorporates a direct comparison with traditional file systems in enterprise GIS. The test carried out using Scott's account is shown in this video https://www.youtube.com/watch?v=zMJV5zV0-zs (accessed on 18 April 2024).

The **test procedure** primarily focused on the time taken and errors in the use of the encryption–decryption solution. Each pair of individuals in the test panel was asked to log in to the ECMSDK shared drive by mapping to it using their laptops. The solution is tested using the classical file-sharing protocol commonly used in enterprise GIS applications. Nevertheless, the model is the same if used by the other protocol servers provided by ECMSDK (that is, web, file transfer, or command line). The user-interaction model is illustrated for two GIS users, Scott and Alan, who require access to single-user and multi-user spatial files. These files must be decrypted for use and encrypted back for protection in storage for a scenario for which both users are concurrently logged in. User Scott has eight single-user spatial files (`kx-site-of-significance-SHP` shapefile dataset) and seven multi-user files (`kx-nz-historic-places-SHP` dataset). User Alan needs access to these seven multi-user files as well as seven additional multi-user spatial files (`lds-nz-bounty-islands` dataset).

**Table 2.** Usability evaluation tasks for two participants using the *Scott* and *Alan* user accounts. Effectiveness focused on user mistakes or the solution's errors made while carrying out these tasks, while efficiency relates to the time taken. The last column refers readers to the figure in Supplementary Materials online that shows the tasks being carried out.

| Task ID | Description | Figure |
|---|---|---|
| 1 | Scott logs in to his ECMSDK home folder using a mapped network drive. | Figures S3 and S4 |
| 2 | Scott notes the *single-* and *multi-user files'* decryption times. | - |
| 3 | Scott ensures the *single-* and *multi-user files* are decrypted and not corrupt. | Figures S5 and S6 |
| 4 | Scott logs out and notes the *single-* and *multi-user files'* encryption times. | - |
| 5 | Scott removes the ".enc" file extension from the *single-user file* for use. | Figure S7 |
| 6 | Alan logs in and checks if the *multi-user files* have been decrypted. | Figure S8 |
| 7 | Scott ensures Alan's login did not corrupt the *multi-user files*. | Figure S9 |
| 8 | Scott, after using the *single-user file*, appends the ".enc" file extension. | Figure S10 |
| 9 | Scott and Alan logout. | Figure S11 |
| 10 | Scott ensures that the *single-* and *multi-user files* are encrypted back. | Figures S12 and S13 |

In this study, the solution's **effectiveness** was measured using data on the number of errors committed by respondents. This includes the solution's inability to perform a task (Tasks 3, 6, 7, and 8). If these numbers is low, the solution already has a good level of effectiveness and vice versa. *Effectiveness* was measured as the degree of task completion. The responses of the test panel on task performance were recorded as either "complete", "complete with errors", or "incomplete", and these were used for assessment. If a respondent's response deviates from the expected one, the reasoning behind his/her completion of the tasks was examined through analysis of the video recordings. Meanwhile, **efficiency** was measured as the time spent to complete a task (Tasks 2 and 4).

To measure user **satisfaction** reliably, a validated psychological questionnaire is generally required, which can consist up to 50 items from categories such as *helpfulness*, *control*, *learn*, and *global* [80]. However, since our solution adds only a single additional step— specification of files to be encrypted or decrypted—to how users work in an enterprise GIS, we therefore believe that such an elaborate questionnaire was not necessary. Instead, we utilised the *Net Promoter Score* [81] metric and asked the respondents one question: *"How would you rate the overall ease of use of our spatial file encryption–decryption solution (on a scale of 1 to 10)?"*

*5.2. Results*

Based on the preliminary study, the spatial file storage security solution was easy to operate for prospective users. The same user interface, that is, a file-sharing client, is used as before, while the only additional task is the specification of files for encryption and decryption. Figures 3 and 4 show two screenshots from the usability testing, while the complete user-interaction model (14 screenshots) is included as part of Supplementary Materials. The screenshots highlight the encryption–decryption aspect, the enforced ECMSDK access control model security and demonstrate how these security aspects are incorporated on top of existing enterprise GIS workflows for increased file protection.

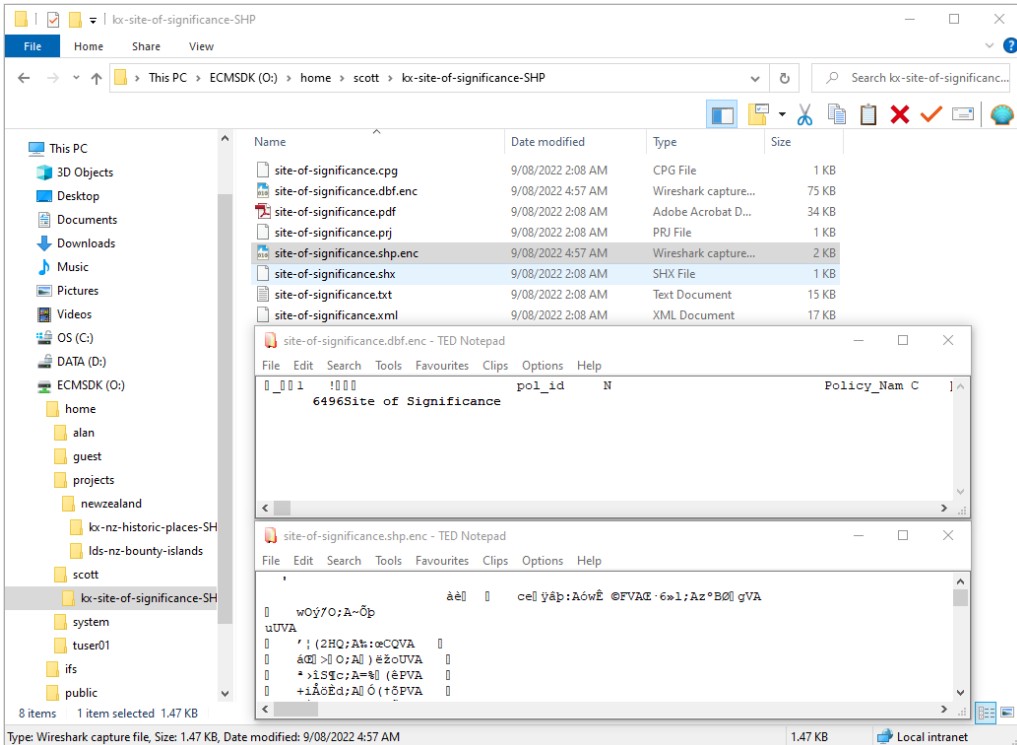

**Figure 3.** Scott's single- and multi-user GIS files are decrypted for use upon his login. ECMSDK ACM security prevents Scott from accessing Alan's home directory.

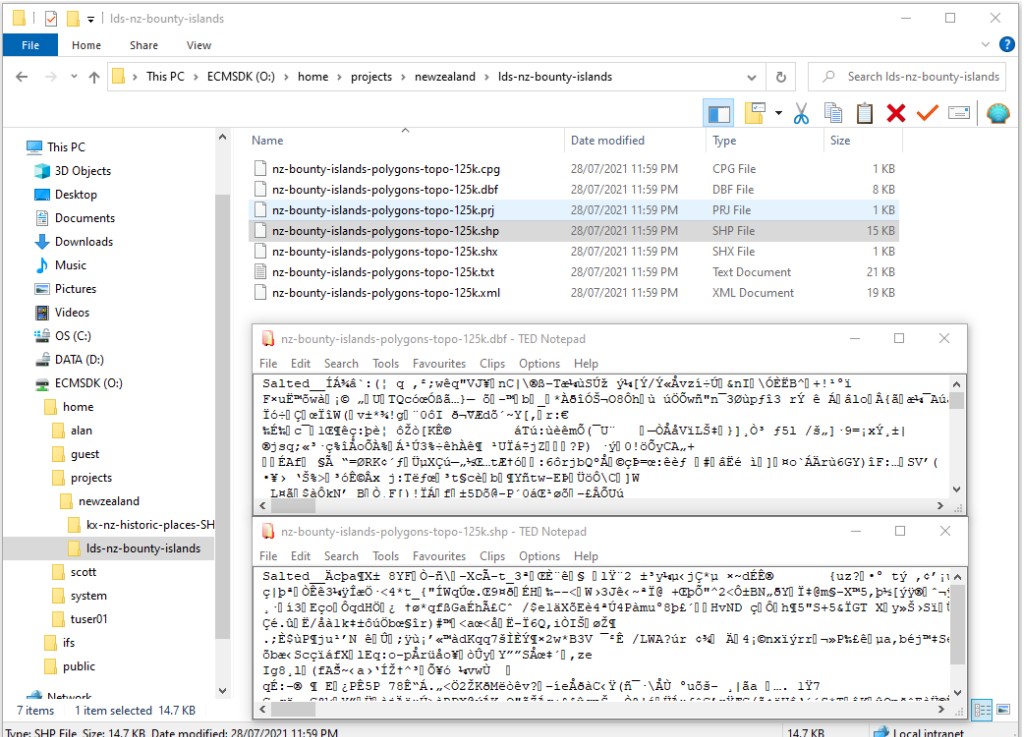

**Figure 4.** Alan's logout encrypts back only seven multi-user files, and the remaining seven multi-user files are unencrypted for Scott, as he still has an active session.

The two aforementioned figures demonstrate how the security solution handles user concurrency. They illustrate how (double) encryption is prevented for multi-user files when a privileged user (Alan) logs out but another privileged user (Scott) still has active sessions. Prevention of (double) decryption when these files have already been decrypted is shown in Figure S9 within Supplementary Materials online.

In usability testing, a low number of errors indicates that the solution is *effective* and vice versa. During testing, no errors from users or the solution were observed when the encryption–decryption solution was used (see Table A1). This is largely due to the solution's simplistic design, and this concludes that the solution is very effective.

*Efficiency* measurement was conducted by comparing the time needed by the respondents to decrypt and encrypt Scott and Alan's *single-* and *multi-user files* in the first and second repetitions. The combined encryption times are presented in Table A2. Since there was no other overhead associated with decryption or encryption, these times are similar to those included in Table 1. The respondents completed the tasks within the expected times in the first repetition since there were no errors.

Our testing determined *user satisfaction* with user responses to one question. The two respondents gave high scores (that is, 9) for their overall ease of use of our security solution (see Table A3). This resulted in a 90% net promoter score, which confirms the high level of user satisfaction.

## 6. Discussion

In meeting our objective, our study proposes a storage security solution that consolidates different types of spatial files with an easy-to-use encryption solution with reasonable encryption times for protection during storage. For a practical deployment of such a security solution, we evaluate its security and usability (including performance).

### 6.1. GIS Security Model and Security Analysis

This section presents a GIS security model based on the proposed security solution for spatial files within a DBFS and discusses each aspect. Our approach aims to address the security vulnerabilities with spatial files within enterprise GIS data *"silos"*. We first propose a *central storage repository* that can store, manage, and control access to spatial files with different formats. A DBFS offers features that are necessary for the future of GIS file management and security that are not provided by traditional file systems. ECMSDK, for instance, provides consolidation for these spatial files into a centralised platform and includes different protocol services commonly used in GIS for accessing and using these files. Essentially, since it is built on top of a multi-purpose database, the underlying database access control mechanism is leveraged by the DBFS providing increased security for files.

The GIS security model is based on the concepts derived from the popular SIS model for security measures [82]. Protecting geospatial data within a DBFS requires a combination of *Technical* and *Administrative security*. Administrative security includes any security measures that focus on managing people and their behaviour. In contrast, technical controls focus on technology. They consist of *IT security*, which pertains to ensuring that computer systems are able to carry out their tasks, and *Physical security* which pertains to controlling access to physical objects and spaces. This includes mechanisms designed to deter unauthorized access to rooms, equipment, documents, and other items. These security measures in this GIS security model would be similar to those employed in enterprise IT systems. The most important characteristic in our study is to ensure the confidentiality of spatial data. For these reasons, here we focus mainly on IT security measures and later highlight the extra emphasis placed on Administrative security.

**IT security** focuses on ensuring these four aspects: *Availability*: providing continual access to data and systems; *Confidentiality*: protecting the privacy of data; *Integrity*: ensuring that data have not been modified; and *Accountability*: holding people responsible for their actions with data and systems.

### 6.1.1. Availability

A DBFS inherits the underlying DBMS's availability, scalability, reliability, and security features that boost data availability for spatial users. For spatial files in enterprise GIS, such a solution promises consolidation and better management while dramatically reducing their vulnerabilities at the storage layer.

### 6.1.2. Confidentiality

*Consolidation* of spatial files into a centralised storage repository does provide attackers with a single point of access to "*more*" valuable data. To this end, our approach rests on the security strength provided by several components: the *access control mechanism* provided by the DBFS, the *cryptographic strength* of the encryption algorithm, how files are *decrypted*, and how encryption keys are *safeguarded*. In our proposed solution, the first security component is access control. Since the DBFS inherits the centralised authentication and access control mechanisms from the underlying DBMS, it provides a strong protection mechanism. The DBFS (ECMSDK) access control model forms the first security layer, where users are first authenticated against their accounts, and based on their privileges, their access is controlled.

The spatial files are further *encrypted* to safeguard against storage-level attacks when the disk or backup removable media is compromised. In our approach, the LOBs storing the original file content are *always in encrypted form*. Upon decryption, temporary files are created, which would be lost upon a server crash caused by an adversary trying to remove the disk. An adversary needs access to the encrypted data, the decryption procedures, and the decryption keys to recover the data. Even if the encrypted data and the decryption algorithm are somehow obtained, the adversary will **not** be able to access the encryption keys stored in a database, which, in turn, are encrypted with another key stored in the *Oracle Wallet*. For sharing these datasets outside the enterprise with individuals having varied access levels, the *MapSafe* geoprivacy tool [44] can be used, which offers a complete approach for safeguarding and verifying geospatial datasets when passed over the network (see Section 2).

Attackers may also attempt to break a targeted cipher through *cryptanalysis*; however, the security provided by the well-researched AES encryption algorithm [83] will prevent such attacks. Another approach would be to carry out a *brute-force* attack with all the possible keys. However, brute-force attacks against spatial files encrypted with the AES algorithm will either take the adversary *longer* to recover the plaintext than the actual value of the data or make it more expensive. In this regard, the longer the key length, the higher the security of the system. This proves that the encryption within the proposed security solution can reasonably stand against cryptanalysis. An alternative method for breaking the encryption focuses on *side-channel* attacks, which attack the physical side effects of its implementation. An error in system design or execution can enable such attacks to succeed. As such, this section presents a substantiation of the security guarantees that the solution offers to enterprise GIS. A malicious adversary can take the following three distinct roles against the security solution:

(1) **Malicious Insider:** The first form of malicious adversary may arise from misbehaving legitimate users. Misbehaving DBFS users' access to spatial files and folders is controlled through a strong access control model. For instance, the ECMSDK product's authentication and authorisation scheme provides a granular and consistent access control mechanism to finely control user access (see end of Section 3.1). This ACM is not weakened by the elevated privileges of OS users using OS protocol servers to access files and prevents users of these accounts from obtaining any useful spatial information, whether encrypted or not.

(2) **Outsider Attackers:** The spatial files within the DBFS can be accessed at multiple instances. This includes once the disk is removed from the DBFS server when files are in (a) *encrypted form* or (b) have been *decrypted for legitimate users*, (c) access of decrypted files on an *unattended PC of legitimate DBFS users*, or (d) access of spatial files when the database

datafiles are *transferred to removable storage media*. The protection provided by the solution to each circumstance is substantiated as follows. Upon removing the disk from the DBFS server when files are in encrypted form or decrypted for legitimate users, the adversary will only have access to the encrypted spatial files stored within the database datafiles. Since the spatial files are decrypted into temporary LOBs, the LOBs containing the original spatial files are never replaced. This feature enables backups to always contain spatial files in encrypted form, and an adversary accessing the backup media will only find encrypted files in all cases.

It leaves the adversary with one option—to break the security of the encryption. If an attacker manages to obtain the encrypted data and decryption algorithm, the adversary will still not be able to access the encryption keys stored in a database, as they are stored encrypted (using another key) inside the Oracle Wallet. Finally, if an adversary chooses to break the encryption security within the solution, the security of the AES algorithm will prevent information recovery.

(3) **DBFS Maintainer:** Finally, an adversary in the form of the DBFS maintainer, who is assigned to perform data backups, transfer them to another place, or destroy them, will always have access to encrypted data only. Since these LOBs are not permanent, they are normally not part of the backup plan and will not be backed up together with the database data while performing online database backups (hot backups). To prevent the maintainer from accessing the table containing the single-user and multi-user cryptographic keys, the `SELECT` access to the table containing the keys can be revoked, which will also prevent the keys from the table from being backed up with the encrypted spatial file.

On the other hand, if the database datafiles are backed up after database shutdown (cold backup), the data warehouse maintainer will again only have access to encrypted datafiles [84]. This is because as soon as the database is closed, all transactions on the LOB are committed, and the decrypted spatial files are encrypted back. The spatial files that were decrypted within temporary LOBs in a temporary tablespace are *freed* as soon as the database is shut down. To decrypt the obtained encrypted files stored in the database backup, the warehouse maintainer needs access to the decryption keys safeguarded using TDE and the Oracle Wallet. To access these keys, he/she needs `SELECT` privilege access to the database tables containing the keys, which are revoked from his default role privilege. Finally, the DBFS maintainer would be faced with needing to break the security of the encryption, which, as previously mentioned, is a very difficult task.

### 6.1.3. Integrity

To verify the originality of the spatial files, users can generate their hash values just before they are encrypted when the user logs off and can compare this with the respective hashes when the files are decrypted the next time they log in.

### 6.1.4. Accountability

Enterprise systems normally include mechanisms to track user actions on sensitive data. Since a DBFS uses the underlying DBMS, database triggers can be coded to execute procedures that track and log any access or changes made to spatial files using `SELECT` or `UPDATE` queries, respectively [85]. In such a manner, even the actions of the DBFS administrator can be tracked.

### 6.2. Administrative Controls

In the GIS security model, more emphasis is now placed on the *DBFS administrator* because of the elevated privileges. The DBFS administrator poses a threat as he/she can access all files and folders within the DBFS. Similarly, the database administrator (DBA) can perform many damaging actions: deleting or replacing the spatial files stored as LOBs in the database table or copying them as soon as they are decrypted. Moreover, the DBA can mimic a privileged user session creation to access the decrypted LOBs in the database table. The DBA can insert a false entry in the `ODMZ_SESSION` table, which would fire the

session-based trigger that would execute the decryption procedure to decrypt the files. Nevertheless, these administrative accounts are by far the most trusted positions in an organisation. To protect valuable data from unwanted disclosure, organisations can limit granting these account privileges to only trusted users.

### 6.3. Usability Analysis

The usability of the security solution is established based on usability testing results.

### 6.3.1. Efficiency (*Including Performance*) Analysis

Our usability testing focused on the *efficiency* of each user's file encryption–decryption performance. Based on the results, the total time required for the encryption or decryption processes was confirmed to be only the respective execution times shown in Table 1. That is to say, the associated ECMSDK user-session-based trigger execution and the subsequent generation of the users' list of spatial files for encryption/decryption are instant. Additionally, since all processing is carried out within the database, there is no additional overhead, regardless of the protocol used to access the spatial files.

In terms of efficiency, the encryption/decryption times for spatial files are small and do enable us to use them for practical implementation in spatial organisations. While it takes more time to encrypt and decrypt spatial files using the DBMS procedures (e.g., Oracle `DBMS_CRYPTO_TOOLKIT`), it takes little time when an encryption executable such as OpenSSL is used. Furthermore, more advanced hardware solutions can further improve performance by an order of magnitude for use with greater ease. Thus, encryption time should not be an issue in enterprise implementation.

Indeed, during normal file operations (opening, editing, and saving spatial files) stored in the DBFS, compared to traditional data storage, there will be delays for a user, as reported in previous research [51,52]. However, these delays were found to be reasonable.

### 6.3.2. Effectiveness and User Satisfaction Analysis

The *effectiveness* of the security solution was confirmed based on no user errors detected during the usability testing. This highlights that there are no weaknesses in the design of the security solution. Similarly, *user satisfaction* was established based on respondents' responses to the single question about the overall ease of use of the encryption solution. Success in both of these aspects was largely due to the security solution's simplistic design.

Our encryption solution makes little changes to how users work with various GIS applications and protocol services in the enterprise. For instance, mapping to a shared drive remains the same, while subsequent decryption and later encryption are carried out automatically in the background. The scheme enables simple specification of single-user and multi-user spatial files that require decryption and encryption when privileged users login and logout from the DBFS, respectively. End users specify single-user files by adding a file extension, which can be performed using any GIS application or protocol service. For specifying users who need access to multi-user spatial files that need encryption, the administrators issue an SQL command to add users to a list. Furthermore, file decryption and encryption processes take place automatically when spatial end-users access and exit the DBFS, respectively, hiding these from the user.

Compared to how users currently work with the various protocol services (e.g., shared folders) in enterprise GIS environments, the only additional task is to specify (single-user and multi-user) files to be encrypted. This step was demonstrated in screenshots: Figures 3 and 4 and those in Supplementary Materials online. Since this specification is the only change and is a small but necessary step for users in any encryption solution, we did not investigate the *user satisfaction* aspect more deeply but confirmed it through responses received from respondents. The ease of specification, together with the automated encryption–decryption model, makes the solution easy to use, potentially resulting in increased uptake.

In this manner, remote end users do not have to use sophisticated software to encrypt and decrypt sensitive files, and the solution is usable on all common protocol services that are used for GIS, which are shipped with ECMSDK. The GIS mapping and spatial analysis subsystems are kept independent of the storage security solution; thus, they require little or no changes to existing applications used in enterprise GIS and how users interact with the solution. Given that the encryption solution overcomes several issues that had previously limited encryption for spatial files, it will encourage its deployment, benefiting the necessary protection of valuable spatial files during storage.

### 6.4. Main Benefits and Limitations

Using a database for file storage and management furnishes several advantages that only a DBMS can offer. As such, this study explored the suitability of a DBFS for spatial file security in enterprise GIS as an alternative to the native filesystem. In doing so, we considered and addressed several challenges. The different scenarios in which spatial files are being used in enterprise GIS led to the design of single- and multi-user encryption–decryption schemes. Moreover, the usability against security conundrum has resulted in a scheme to carry these functions out automatically, hiding them from the users. Also, we propose schemes to address the challenges in an encryption solution implementation, namely, automation, user concurrency, and a protection scheme for preventing access to decrypted data. Finally, the security solution brings minimal changes to how users work with spatial files in enterprise GIS. Users only have to specify the files that need encryption–decryption. We have substantiated that using a DBFS for spatial file storage is slower than a native file system but better in terms of the overall security of spatial files in enterprise GIS.

The main benefits and limitations of the proposed solution are as follows. Many GIS organisations are storing their datasets as sets of files and folders—this can be a more convenient way to share datasets within an organisation compared to a *DBMS*. The latest desktop GIS applications (such as *ESRI ArcGIS Pro*) do not address the security solution at all to work with spatial files and personal geodatabase workspaces. The existing security solutions for spatial datasets mentioned in Section 2 do not provide varied ways to store and use spatial datasets for organisations dealing with large spatial datasets. Existing solutions apply only to certain types of spatial data and are mainly concerned with encrypting spatial files without a usable encryption solution around them.

To this end, the proposed solution for sensitive spatial files offers comprehensive multi-level security (authentication, encryption, and authorisation) together with an automated and usable encryption–decryption solution. In addition, there is no additional cost associated with using this approach. The DBFS product used in our investigation—*ECMSDK*—is available as an open-source product. Indeed, the underlying Oracle XE database we used has size limitations. Instead, the *PostGreSQL* database can be used, and as such, no additional cost will be required to deploy the security solution. However, re-writing of all *Oracle PL/SQL* scripts to *Postgres PL/pgSQL* would be necessary.

The main limitation of the proposed solution is that since DBFS adds a database layer between the GIS applications and the file system, more time is required when users work with spatial files: that is, opening, editing, and saving files [51,52]. Additional time is also required during file encryption and decryption. Both of these additional times, however, are a fair trade-off considering the spatial file management and security features offered by the proposed security solution. Another limitation is the elevated privileges of the DBFS and DBMS administrators, who both can access the files in the database once they have been decrypted during users' sessions. To protect against unwanted disclosure, organisations can limit granting these administrative account privileges to only trusted individuals.

Consolidation does pose a *single point of failure*. However, while permitting storage of spatial files of different formats into a centralised repository, the DBFS prevents a single point of failure to a significant level by taking advantage of the availability, scalability, and security features of the underlying DBMS, with the further possibility of deploying multiple *DBFS nodes* [48]. These features are comparable with cloud-based file storage,

and they extend current file systems, enabling GIS organisations to deploy a robust in-house file storage platform. Moreover, spatial files are protected in storage via encryption.

However, as part of future research, we will explore a complete *decentralised* architecture wherein multiple DBFS nodes protect spatial files in-house. In this completely decentralised architecture, there would be no central server and thus no *single point of failure*. A multi-node ECMSDK architecture currently requires a master–slave configuration, for which a dedicated master node must always be online. To this end, the implementation of a complete *peer-to-peer* (P2P) architecture with multiple ECMSDK nodes can be undertaken, where—similar to *IPFS*—a spatial file uploaded to any node would first be encrypted and then synced with all other ECMSDK nodes. This approach would leverage the DBFS file management features and encryption–decryption model at each node in the P2P architecture within an enterprise GIS. To the best of our knowledge, no such peer–peer architecture based on DBFSs is currently available in the market.

## 7. Conclusions

The increased use and sharing of spatial data at a time when cloud-based data storage has become a norm has raised security and privacy concerns from both international regulators and from users themselves. However, the implementation of on-premise, secure, and yet usable systems is a challenge for software and service providers. Addressing this challenge requires actively integrating existing security components in a way that revitalises localised storage.

A file system built on a general-purpose database provides GIS organisations with several essential features not offered by traditional file systems. Spatial files with different formats can be consolidated in a central repository that takes advantage of the DBMS's availability, scalability, and reliability features and enables searching, content management, and a sound access control model built on the database. An additional encryption layer provides an easy-to-use interaction model that is an improvement over any complicated cryptographic processes, overcoming several issues that had previously prevented users from using a real encryption solution for protecting spatial files within enterprise GIS.

As such, the security solution furnishes a consolidated and secure platform for storing and managing spatial data within the enterprise. These are essential features for informed decision-making and can help organisations with complying with data privacy laws and regulations while protecting them from legal liability and fines. As more GIS organisations realise the benefits of storing and managing spatial files inside the database, our approach positions itself as an ideal encryption model for protecting these files, drawing more attention to this solution.

The debate over the performance and security of the proposed solution will continue to be a determining factor for its deployment. Nevertheless, one thing is clear: the solution does provide the essential storage infrastructure that will significantly reduce security vulnerabilities with stored spatial files. Essential GIS services will be able to keep their existing workflow intact without any alteration to how users work in an enterprise collaborative environment and with maximum security for stored spatial data. This level of convenience provided by the encryption solution will promote its use among users and, as a result, protect valuable spatial files throughout enterprise GIS (even in cases of compromise).

**Supplementary Materials:** All supplementary materials referred to in this article are available online within this GitHub repository https://github.com/sharmapn/DBFSFileCrypto.

**Author Contributions:** Conceptualization, P.S. and M.G.; methodology, P.S.; software, P.S.; validation, M.M. and M.G.; formal analysis, M.M.; investigation, P.S.; writing—original draft preparation, P.S.; writing—review and editing, M.G. and M.M.; supervision, M.G. and M.M. All authors have read and agreed to the published version of the manuscript.

**Funding:** This research received no external funding.

**Informed Consent Statement:** This paper does not contain any studies with human participants or animals performed by any of the authors.

**Data Availability Statement:** There were no new datasets generated during the current study.

**Conflicts of Interest:** The authors have no conflicts of interest to declare that are relevant to the content of this paper.

### Appendix A

This appendix includes the results of user testing.

**Table A1.** Average number of errors made in tasks for each repetition.

| Repetition | Average Number of Errors Made in Tasks | | | | | | | | | |
|---|---|---|---|---|---|---|---|---|---|---|
| | Task 1 | Task 2 | Task 3 | Task 4 | Task 5 | Task 6 | Task 7 | Task 8 | Task 9 | Task 10 |
| 1 | 0 | 0 | 0 | 0 | 0 | 0 | 0 | 0 | 0 | 0 |
| 2 | 0 | 0 | 0 | 0 | 0 | 0 | 0 | 0 | 0 | 0 |
| 3 | 0 | 0 | 0 | 0 | 0 | 0 | 0 | 0 | 0 | 0 |
| 4 | 0 | 0 | 0 | 0 | 0 | 0 | 0 | 0 | 0 | 0 |

**Table A2.** Average encryption time per task repetition.

| Efficiency | Average (in Minutes) | Efficiency | Average (in Minutes) |
|---|---|---|---|
| Repetition 1 | 2.711 | Repetition 3 | 2.221 |
| Repetition 2 | 2.506 | Repetition 4 | 2.103 |

**Table A3.** User satisfaction question.

| Question | Respondents | |
|---|---|---|
| | 1 | 2 |
| How would you rate the overall ease of use of the spatial file encryption–decryption solution *(from 1–10)*? | 9 | 9 |

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
