# Peer review of "A Usable Encryption Solution for File-Based Geospatial Data within a Database File System"

_jcp, doi:10.3390/jcp4020015_

Round 1

Reviewer 1 Report

The goal of this paper is to provide a secure solution for GIS data protection. In general, the introductory part seems to be very well argued except for the part on usability. The "usable security" aspect is mentioned, but the authors do not specifically address the issue by discussing relevant literature (I suggest making reference to recent reviews of the literature, like the most recent "Usable Security: A Systematic Literature Review" published in  Information by Di Nocera, Tempestini, and Orsini).

The authors claim that they want to identify a security solution that is at the same time also usable, but there is no empirical study supporting the actual usability of their solution. Showing two screenshots cannot substitute for usability evaluation. 

For usability evaluation in their field, a reference could be made to: "Spatial Metadata Usability Evaluation" published in ISPRS International Journal of Geo-Information by Kalantari and colleagues.

My request for the authors is to add a usability evaluation of their solution using the methodology they prefer. For example, heuristic evaluation should be fast enough to implement (https://www.nngroup.com/articles/how-to-conduct-a-heuristic-evaluation/) and could add substance to their claim that the solution is usable compared to other solutions.

GIS should be spelled out and briefly described at the beginning of the paper for the reader who knows nothing about it.

What does usability mean in this study? The authors seem to refer more to ease of use than anything else. It would be good to include a clear definition of usability (perhaps in line 193). Refer to ISO 9241-11:2018 for the definition of usability. 

Author Response

We have attached our responses to the PDF document.

Reviewer 2 Report

1. Refine graph titles and labels for clarity, ensuring accessibility for diverse audiences. Adjust color schemes and add a brief narrative for better interpretation. 2. Clearly outline the research problem's significance, provide additional context on existing literature, and integrate a captivating hook to engage readers from the outset. 3. Maintain a consistent and professional formatting style throughout the presentation. Use visual elements purposefully to avoid clutter and ensure audience engagement. 4. Expand on implications and offer a nuanced interpretation of results, acknowledging limitations. Articulate the practical relevance of the study and its contributions to the field. 5. Reference format must be uniform. Please cite several papers from the literature on detection. a. CorrAUC: a Malicious Bot-IoT Traffic Detection Method in IoT Network Using Machine Learning Techniques.  6. Summarize key contributions concisely, emphasizing the study's significance. End with a forward-looking statement, indicating potential future research directions.

1. Refine graph titles and labels for clarity, ensuring accessibility for diverse audiences. Adjust color schemes and add a brief narrative for better interpretation. 2. Clearly outline the research problem's significance, provide additional context on existing literature, and integrate a captivating hook to engage readers from the outset. 3. Maintain a consistent and professional formatting style throughout the presentation. Use visual elements purposefully to avoid clutter and ensure audience engagement. 4. Expand on implications and offer a nuanced interpretation of results, acknowledging limitations. Articulate the practical relevance of the study and its contributions to the field. 5. Reference format must be uniform. Please cite several papers from the literature on detection. a. CorrAUC: a Malicious Bot-IoT Traffic Detection Method in IoT Network Using Machine Learning Techniques. 6. Summarize key contributions concisely, emphasizing the study's significance. End with a forward-looking statement, indicating potential future research directions.

Author Response

(The authors gave the same response as above.)

Reviewer 3 Report

This paper presents a user-centred and reasonably efficient security solution including encryption, decryption, and automation within collaborative enterprise GIS to keep this data secure.  

However, the paper needs further revision to meet the quality requirements of the journal. 

Detail comments are as follows:

1. The full name of the abbreviation, including but not limited to GIS and AES needs to be stated in the abstract.

2. While the article provides a detailed explanation of the engineering aspects of the proposed method, it lacks sufficient coverage of the core algorithm, mathematical formulas, and scientific experimental design, which results in a lack of focus on the scientific questions.

3.  I would suggest the author consider incorporating quantifiable metrics in the performance analysis and other results analysis sections.

4. For a general research paper, it is important to avoid an excessive number of references and ensure that all cited literature is highly relevant. I recommend reviewing the references to check for any citations that may have low relevance to the topic at hand. 

Author Response

(The authors gave the same response as above.)

Round 2

Reviewer 1 Report

The authors tried to improve their manuscript by accommodating my requests concerning usability. However, I still feel that usability has not been thoroughly addressed. From what I understand, only two subjects performed tasks, which qualifies the study more as a single-user study than an accurate usability study. Perhaps the authors could clarify this by reporting that this is not a proper usability study but "Non-Technical Single User Testing" (see https://usabilitygeek.com/what-is-the-mum-test/). Relative to the satisfaction measure, if a single-item measure were really to be used, it would have been preferable to use the Net Promoter Score (https://en.wikipedia.org/wiki/Net_promoter_score). Moreover, in this case, it is reported that there are four respondents. I have serious difficulty understanding how these tests were conducted and on whom.

The method section must include a detailed description of the users involved and the steps they followed.

Author Response

Dear Reviewer,
We would like to thank your review on the manuscript.
We have addressed your comments. Our responses are included in blue text, while our additions to the manuscript are shown in green text.
Our changes to the manuscript have been highlighted in yellow text.

Yours sincerely,
Pankaj, Michael, and Michael
